# Detecting Systematic Weaknesses in Vision Models along Predefined Human-Understandable Dimensions

**Sujan Sai Gannamaneni**                                    *sujan.sai.gannamaneni@iais.fraunhofer.de*
*Fraunhofer IAIS, Lamarr Institute*

**Rohil Prakash Rao**                                    *rohil.prakash.rao@iais.fraunhofer.de*
*Fraunhofer IAIS*

**Michael Mock**                                    *michael.mock@iais.fraunhofer.de*
*Fraunhofer IAIS*

**Maram Akila**                                    *maram.akila@iais.fraunhofer.de*
*Fraunhofer IAIS, Lamarr Institute*

**Stefan Wrobel**                                    *stefan.wrobel@iais.fraunhofer.de*
*Fraunhofer IAIS, University of Bonn*

**Reviewed on OpenReview:** *https://openreview.net/forum?id=yK9pvt4nBX*

## Abstract

Slice discovery methods (SDMs) are prominent algorithms for finding systematic weaknesses in DNNs. They identify top-k semantically coherent slices/subsets of data where a DNN-under-test has low performance. For being directly useful, slices should be aligned with human-understandable and relevant dimensions, which, for example, are defined by safety and domain experts as part of the operational design domain (ODD). While SDMs can be applied effectively on structured data, their application on image data is complicated by the lack of semantic metadata. To address these issues, we present an algorithm that combines foundation models for zero-shot image classification to generate semantic metadata with methods for combinatorial search to find systematic weaknesses in images. In contrast to existing approaches, ours identifies weak slices that are in line with predefined human-understandable dimensions. As the algorithm includes foundation models, its intermediate and final results may not always be exact. Therefore, we include an approach to address the impact of noisy metadata. We validate our algorithm on both synthetic and real-world datasets, demonstrating its ability to recover human-understandable systematic weaknesses. Furthermore, using our approach, we identify systematic weaknesses of multiple pre-trained and publicly available state-of-the-art computer vision DNNs.

## 1 Introduction

With recent advances in machine learning (ML), there has been a significant improvement in the modeling of unstructured data, such as images. However, for safety-critical applications, ML models need to be developed with a focus on trustworthiness by investigating and correcting potential failure modes. To that end, systematic errors of DNNs need to be studied and rectified. Hidden stratification (Oakden-Rayner et al., 2020) and fairness-related bias (Buolamwini & Gebru, 2018; Wang et al., 2020; Li et al., 2023) due to spurious correlations (Xiao et al., 2021; Geirhos et al., 2020; Mahmood et al., 2021) and underrepresented subpopulations (Sagawa* et al., 2020; Santurkar et al., 2021) are some examples of potential failure modes where the error or weakness is systematic in nature. The existence of these modes implies that there are

slices [1] of data where the performance of the DNN-under-test (**DuT**) is worse than the average performance on the entire test dataset. Although identifying slices with weak performance would be trivial by simply grouping samples on which models have high error, identifying slices that are both semantically coherent and have high error is challenging. This is due to the lack of semantic metadata that describes the slices for many data domains (e.g., images, text). Despite this challenge, identifying such slices provides a human-understandable global explanation of the model behavior. Moreover, semantically coherent weak slices offer actionable insights for debugging and auditing models.

From a safety and certification perspective, upcoming standards (e.g., ISO/PAS 8800 (ISO, 2024)), and works with a focus on AI in automotive (Koopman & Fratrik, 2019; Burton et al., 2022), aerospace (European Union Aviation Safety Agency, 2024) and railway (Zeller et al., 2023) domains have highlighted the importance of data completeness and quality using, in most cases, Operational Design Domains (ODDs). In automotive, Herrmann et al. (2022) have proposed ontologies for different traffic participants that can be used to build ODDs for automated driving. The goal of using such ODDs is to describe the scope of AI applications in terms of human-understandable, safety-relevant dimensions where comprehensible safety argumentations can be built w.r.t. robustness, explainability, and interpretability. To facilitate building such safety augmentations, testing approaches for ML developers and safety experts that evaluate DNN performance and identify systematic weaknesses are essential.

Although in recent years, several works (Chung et al., 2019; Sagadeeva & Boehm, 2021; d'Eon et al., 2022; Eyuboglu et al., 2022; Metzen et al., 2023; Plumb et al., 2023; Jain et al., 2023; Gao et al., 2023; Chen et al., 2023) have proposed methods for analyzing systematic weaknesses, there is a lack of focus on identifying weaknesses of models evaluated on real-world datasets, where the weaknesses align with human-understandable semantic concepts defined, for example, by safety experts in ODDs. We argue that it is more beneficial from a safety perspective if the approaches to identify systematic weaknesses are ODD-compliant for two main reasons: (i) the slices are **useful** as the identified vulnerabilities are aligned with human-understandable safety-relevant dimensions, and (ii) the slices are **actionable** as ML developers can gather more data to retrain or reweight existing samples to improve performance along the safety-relevant dimensions. We address the challenge of analyzing unstructured image data by designing an algorithm that leverages recent advances in foundation models and systematic weakness analysis methods for structured data. Our contributions can be summarized as follows:

- We introduce an algorithm that takes in an image dataset, ODD description, and performance values of a **DuT** as inputs and outputs systematic weaknesses of the **DuT** (see section 3). Our implementation is available at `https://github.com/sujan-sai-g/Systematic-Weakness-Detection`.

- Concretely, as part of the metadata generation module, we make use of CLIP (Radford et al., 2021) to leverage its rich joint image, text embedding space. As part of the slice discovery module, we propose using SliceLine (Sagadeeva & Boehm, 2021) with modifications to identify weak slices that align with the ODD (see section 3).

- In addition, we address the noisy nature of metadata generation and propose a way to recover relevant weak slices even if CLIP labeling is suboptimal. We empirically evaluate the behavior of our algorithm at various levels of label quality using synthetic data (see section 4).

- Furthermore, we evaluate multiple pre-trained and publicly available **DuTs** using our algorithm on real-world datasets and provide insights into their systematic weaknesses (see section 5).

## 2 Related Work

In this section, we review the recent progress in analyzing systematic weaknesses using slice discovery methods (SDMs) (Eyuboglu et al., 2022) for structured and unstructured data and highlight their connection to interpretability and feature attribution methods.

---

[1]In the literature, slices are often also called subgroups or subsets of data. All three terms are used interchangeably in relation to systematic weakness analysis.

For structured data, methods such as SliceFinder (Chung et al., 2019) and SliceLine (Sagadeeva & Boehm, 2021) leverage the rich metadata available in the form of features to slice the data and exhaustively search for top-k low-performing slices. The differences between these two approaches lie in the scoring of errors, the pruning strategy, and how they handle slice sizes. Although these two approaches were explicitly developed to identify systematic weaknesses, subgroup-discovery techniques (Atzmueller, 2015), a subset of data mining, have a similar problem formulation and could also potentially be used for slice discovery of structured data.

For unstructured data such as images, where metadata is not directly available, SOTA approaches have taken two lines of research. In the first line of prior work, referred to as *slice-and-tag* approaches by (Chen et al., 2025), for a given test dataset, DNN embeddings are used as proxies for coherency. Weak-performing slices of the data are obtained by clustering these embeddings along with model errors. Here, approaches such as Spotlight (d'Eon et al., 2022) perform clustering on the embeddings of the final layers of the **DuT** itself. In contrast, recent approaches leverage the joint embedding space of foundation models such as CLIP (Radford et al., 2021) and apply Gaussian mixture models like in DOMINO (Eyuboglu et al., 2022) or SVMs like in SVM-FD (Jain et al., 2023) to identify coherent clusters. In Spotlight, an additional step involving humans is required to inspect and understand what uniquely constitutes a weak slice. DOMINO and SVM-FD automate the slice description process to reduce human effort and bias using an additional DNN. In all these approaches, as coherence is only loosely enforced based on DNN embeddings, it is not always clear what specific human-understandable concept uniquely constitutes a slice. Without this knowledge, it would be unclear to the ML developers what new data samples would need to be collected to retrain the model and fix the systematic weakness. To mitigate this problem, some approaches (Gao et al., 2023; Slyman et al., 2023) propose iterative human-in-the-loop testing to ensure that the identified slices are human-understandable.

In the second line of prior work, referred to as *tag-and-slice* approaches by (Chen et al., 2025), inspired by counterfactuals and leveraging CLIP, several approaches (Wiles et al., 2022; Metzen et al., 2023) propose synthetically generating new (counterfactual) images that would lead to erroneous predictions by controlling the content and data shift in the image. Among these, PromptAttack (Metzen et al., 2023) also proposes to identify weaknesses that are aligned with the ODDs. However, while PromptAttack generates new samples using image-generation DNNs, which could potentially introduce biases due to domain shift, our approach is more closely aligned with earlier methods that evaluate a DNN on a given test dataset. In this direction, HiBug (Chen et al., 2023) utilizes a GPT-based model to assign attributes to a given dataset. Building on this and appearing concurrently with our work, HiBug2 (Chen et al., 2025) extends HiBug with a search algorithm to identify weak slices. While we also apply attributes to the data to perform a subsequent weak slice search, we, instead, opt for the less compute-intensive CLIP (Radford et al., 2021) model to generate attributes. Additionally, we develop a Bayesian framework to compensate for the label noise that occurs from the attribution.

In contrast to SDMs, local interpretability and feature attribution methods (Ribeiro et al., 2016; Lundberg & Lee, 2017), while linking achieved understandability to actionability (Guidotti et al., 2022), identify local explanations and not the global systematic weaknesses. In addition, the feature attribution methods themselves might not always be robust or consistent (Krishna et al., 2024).

## 3 Method

In this section, we present our algorithm for weakness detection on the basis of human-understandable semantic dimensions. To this end, we introduce notation regarding metadata and slicing, discuss the generation of metadata, formulate DNN weakness within a Bayesian framework to account for the impact of noise, and lastly detail how such impact can be acknowledged within slice discovery algorithms.

**Notation**: Consider a DNN-under-test (**DuT**) $M$ trained on some computer vision task. Let $\mathcal{D}$ be the (test) dataset containing the inputs and the corresponding task-related ground truth. For each sample $s_i \in \mathcal{D}$, using some per-sample performance metric (e.g., intersection over union (IoU)) and, if applicable, applying some threshold, we obtain binarized **DuT** errors, defined as $e_i \in \{0, 1\}$. Each sample is either correctly ($e_i = 0$) or incorrectly ($e_i = 1$) predicted by the **DuT**. Here, we slightly deviate from conventional notation by considering individual samples rather than the **DuT** inputs. Although identical for image classification, in the case of object detection, multiple samples (i.e., objects) may be present in a given input image, over

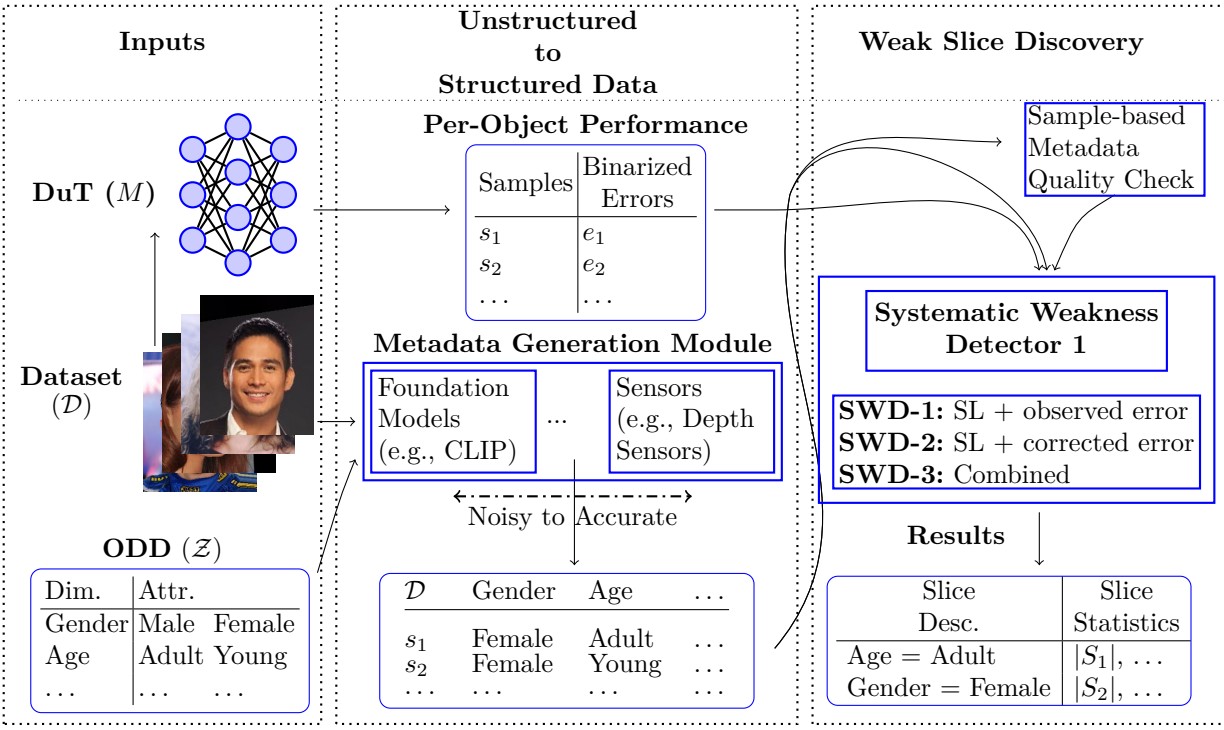

Figure 1: Our algorithm for finding systematic weaknesses of CV models. Given a model, a test dataset, and an ODD description for the objects we are interested in, we build a database of object-level performance and metadata in a structured format. Weak slice discovery methods are then applied to this database to identify top-k weak slices of the model.

which inference is performed. Using a set of samples with individual errors $e_i$ allows us to define slices $\mathcal{S} \subseteq \mathcal{D}$ of the data and their corresponding (average) error rate $\bar{e}|_{\mathcal{S}}$, defined as $\frac{1}{|\mathcal{S}|} \sum_{s \in \mathcal{S}} e_s$. One of the goals of slice discovery methods is to find slices where $\bar{e}|_{\mathcal{S}}$ is significantly worse than the global average $\bar{e}|_{\mathcal{D}}$. However, this constraint alone could be trivially satisfied by selecting all samples where $e_i = 1$. But, this, in general, would reveal no further information than the known data-points with bad performance.

As outlined in section 2, existing works in slice discovery can be broadly categorized into two areas of research: (i) *slice-and-tag* and (ii) *tag-and-slice*, as introduced by (Chen et al., 2025). The first category includes methods such as DOMINO (Eyuboglu et al., 2022), Spotlight (d'Eon et al., 2022), and SVM-FD (Jain et al., 2023). These approaches begin by embedding each input $s$ using some DNN $\mathcal{E}$ (e.g., CLIP or the **DuT** itself). In the resulting embedding space, clustering or classification approaches (e.g., using GMMs or SVMs) are applied to the joint space of **DuT** performance $e$ and the embeddings $\mathcal{E}(s)$ across the full test dataset. (Plumb et al., 2023) offers a helpful overview of the different embedding and clustering approaches used in slice discovery. This constitutes the slicing step with the expectation that the obtained slices $\mathcal{S} \subseteq \mathcal{D}$ retain semantic coherency and also contain high **DuT** errors. Next, as part of the tagging step, some approaches use language models to describe $\mathcal{S}$. As the approaches do not explicitly enforce the descriptions to follow a set of semantics within $\mathcal{Z}$, the descriptions might be very general. Moreover, there is no guarantee that a coherent region in the latent space corresponds to a single, human-understandable concept. Instead, mixtures across broader categories may arise, and elements of an assumed semantic concept for $\mathcal{S}$ may also be reflected in data points beyond the slice. In contrast, the *tag-and-slice* approaches like ours and HiBug2 (Chen et al., 2025) focus on "tagging" each input $s$ from a predefined set $\mathcal{Z}$ of semantic dimensions and corresponding attributes. To enable this, some DNN $\mathcal{G}$ (e.g., CLIP or GPT) is used to both embed and classify each input $s$ based on the dimensions in $\mathcal{Z}$. For instance, in pedestrian detection, such dimensions can be fairness-related, such as "gender" or "age", with attributes like "young" or "old", but may also include other safety-relevant aspects such as "occlusion" or "clothing-color". This way, unstructured

image data is transformed into structured metadata containing the relevant semantic dimensions. As part of the slicing step, this metadata is combined with the **DuT** errors $e$ and analysed using a search algorithm. These approaches offer direct interpretability of the slices, and the choice of dimensions can be aligned, e.g., with existing safety considerations. More concretely, larger parts of Operational Design Domains can often be formalized in terms of simple sets $\mathcal{Z}$. For our work, the concrete set used is inspired by the ODD within Herrmann et al. (2022) from the autonomous driving domain and we use $\mathcal{Z}$ and ODD synonymously in the remainder.[2] PromptAttack (Metzen et al., 2023) also falls within the *tag-and-slice* category. However, in this method, instead of metadata generation for existing data, new data points $s'$ are generated that are not within $\mathcal{D}$ using diffusion models based on semantics defined in $\mathcal{Z}$. By using the generated $s'$ as a test set, weaknesses of **DuT** are evaluated. Here, issues related to syn2real gap and systematic weaknesses of diffusion models must be considered when evaluating the results.

In fig. 1, we present our algorithm, where, using inputs such as a dataset $\mathcal{D}$, a predefined ODD $\mathcal{Z}$, and a **DuT** $M$, we transform the task of finding systematic weaknesses in the unstructured data domain into a problem in the structured data domain. The algorithm is designed with modular components for adaptability. The first module handles the generation of structured metadata, while the second module applies the weak slice search algorithm to the generated structured metadata. With a structured description of the data, we can formulate slices as rules over $\mathcal{Z}$, e.g., gender $=$ male $\wedge$ occlusion $= (0.9, 1.0]$. This allows for a more probabilistic notation $p(e|\mathcal{S})$ of the true error given the slice. Slice discovery is then the task of finding (coherent) conditions $\mathcal{S}$ such that the conditional expectation is maximized.

**Metadata Generation**: While there is great interest from safety experts and certification bodies in using ODDs for safety argumentation, metadata that align with the ODD are scarcely available for most, particularly image, domains. Human annotation of such metadata is often out of scope for large datasets due to cost and time constraints. However, an automated metadata generation module that captures different semantic dimensions of $\mathcal{Z}$ is feasible with existing technologies. For example, a multi-modal foundation model like CLIP with its joint image and text embedding space could be a potential candidate for such automated annotation out of the box or after fine-tuning. Inspired by Gannamaneni et al. (2023), we leverage CLIP to generate metadata for each sample $s$ in our dataset across all dimensions in $\mathcal{Z}$ using the embedding and classification function $\mathcal{G}$. Specifically, we first obtain the embedding of an input sample $s$ using the CLIP vision encoder. Simultaneously, we generate embeddings for attribute-specific prompts using the CLIP text encoder, following the ensemble prompting strategy introduced by Gannamaneni et al. (2023), which builds upon the ensemble prompting technique proposed in Radford et al. (2021). For example, in the case of the *gender* dimension with attributes such as "male" and "female", we manually construct ensemble prompts for each attribute and encode them using the text encoder. Attribute classification for each dimension is performed by computing similarity scores between the image and text embeddings; the attribute with the highest score is assigned. For example, if sample $s$ depicts a "male," the corresponding similarity score would be highest, and "male" would be assigned as the metadata value for the *gender* dimension for that sample. This procedure facilitates the extraction of structured metadata from unstructured image data.

Taking the ontology for pedestrians from the automotive domain as a baseline, a qualitative evaluation of CLIP's capability was performed by Gannamaneni et al. (2023). While CLIP achieved SOTA level zero-shot performance on different dimensions such as gender, skin-color, and age for portrait shots of human faces in the CelebA (Liu et al., 2015) dataset, they observed a drop in performance on real-world datasets containing pedestrians like in the Cityscapes (Cordts et al., 2016) dataset. The drop in performance can be attributed to more challenging conditions, such as complex poses, low illumination, and high occlusion. These observations, along with our experiments, show that the function $\mathcal{G}$ is subject to varying degrees and types of uncertainty, depending on the dimensions of $\mathcal{Z}$: (i) the presence of data-based (aleatoric) uncertainties, i.e., where the image resolution is low or the object in question is heavily occluded or distant, leading to errors in the generated metadata. (ii) the presence of model-based (epistemic) uncertainties, i.e., where the function $\mathcal{G}$ exhibits suboptimal performance. While (i) can occur in the case of both human and CLIP-based

---

[2]In practice, ODDs are often ontological in nature, and only sub-components of them might be captured by semantic sets of the form $\mathcal{Z}$. For example, "trucks" and "pedestrians" could be part of the same ODD for a **DuT** but would have different semantic dimensions.

annotation, (ii) occurs more prominently in non-human, automated labeling.[3] Therefore, any method that aims to consider metadata generated using such techniques should take into account the incurred noise in downstream tasks.

**Bayesian Framework to Account for the Impact of Noise**: To address the uncertain nature of classification, we extend the previous slice notation of the error to the joint probability $p(e, \mathcal{C}, \mathcal{S})$, where $\mathcal{C}$ represents the outcome of automated labeling for some attribute of a dimension, while $\mathcal{S}$ denotes the corresponding ground truth. For simplicity, we drop the indices and make the additional assumption that $\mathcal{S}, \mathcal{C}$ can be seen as binary, i.e. they may either be true ($\mathcal{S}$, $\mathcal{C}$) or not true ($\neg\mathcal{S}$, $\neg\mathcal{C}$), respectively (for details on the non-binary case, see appendix B.4). Using Bayes' Theorem and marginalizing over $\mathcal{C}$ or $\mathcal{S}$, we can express

$$p(e|\mathcal{S}) = p(e|\mathcal{C}, \mathcal{S})r_{\mathcal{C}} + p(e|\neg\mathcal{C}, \mathcal{S})\left(1 - r_{\mathcal{C}}\right), \tag{1}$$

$$p(e|\mathcal{C}) = p(e|\mathcal{C}, \mathcal{S})p_{\mathcal{C}} + p(e|\mathcal{C}, \neg\mathcal{S})\left(1 - p_{\mathcal{C}}\right). \tag{2}$$

Here, $p(e|\mathcal{S})$ represents the true slice error, while $p(e|\mathcal{C})$ denotes the observed slice error. Furthermore, $p_{\mathcal{C}} = p(\mathcal{S}|\mathcal{C})$ and $r_{\mathcal{C}} = p(\mathcal{C}|\mathcal{S})$ are shorthand for precision and recall of the labeling function $\mathcal{G}$ measured towards the ground truth, and are used in our algorithm, fig. 1, for the quality check. A detailed derivation of the equations is provided in appendix B.1. Making these relations explicit allows us to investigate the hypothesis typically underlying Slice Discovery Methods in more detail. Specifically, based solely on the observed slice performance/weakness $p(e|\mathcal{C})$, one may conclude that a related data property $\mathcal{S}$ represents a weakness of the **DuT**, i.e., we assume that $p(e|\mathcal{S})$ also has a comparable performance/weakness. While in our algorithm the relation between $\mathcal{S}$ and $\mathcal{C}$ is explicit as the latter is given by a classifier for the former, in other approaches (d'Eon et al., 2022; Eyuboglu et al., 2022; Jain et al., 2023) the relation is implicit, as observed sets $\mathcal{C}$ are interpreted to indicate a meaning of $\mathcal{S}$ (typically referred to as a slice label). Another assumption typically made is the independence between the labeling function $\mathcal{G}$ and **DuT**. This independence would imply that the errors of the DuT do not depend on the noise (errors) of $\mathcal{G}$. Specifically, for a semantic attribute, the error rates $p(e|\mathcal{C}, \mathcal{S})$ when $\mathcal{G}$ is correct and the error rate $p(e|\neg\mathcal{C}, \mathcal{S})$ when it is not should be (approximately) equal. However, our experiments indicate that this is not always the case; therefore, we denote the difference by

$$\delta p(e|\mathcal{S}) = p(e|\neg\mathcal{C}, \mathcal{S}) - p(e|\mathcal{C}, \mathcal{S}). \tag{3}$$

Please note that $\delta p$ describes intra-set variances of the error rate in the set $\mathcal{S}$ and is not a conditional probability on its own. Taking into account this potential dependence, we can derive the true error from the observed error exactly given the performance of the annotation process using

$$p(e|\mathcal{S}) = \underbrace{\frac{p(e|\mathcal{C})\,p_{\neg\mathcal{C}} + p(e|\neg\mathcal{C})\,(p_{\mathcal{C}} - 1)}{p_{\mathcal{C}} + p_{\neg\mathcal{C}} - 1}}_{\text{independence assumption}} + \underbrace{\delta p(e|\mathcal{S})\overbrace{\left(\frac{p_{\mathcal{C}}p_{\neg\mathcal{C}}}{p_{\mathcal{C}} + p_{\neg\mathcal{C}} - 1} - r_{\mathcal{C}}\right)}^{\kappa_{\mathcal{S}}} + \delta p(e|\neg\mathcal{S})\overbrace{\frac{(p_{\mathcal{C}} - 1)p_{\neg\mathcal{C}}}{p_{\mathcal{C}} + p_{\neg\mathcal{C}} - 1}}^{\kappa_{\neg\mathcal{S}}}}_{\text{correction terms}}. \tag{4}$$

As long as the independence assumption is (approximately) valid, implying $\delta p(e|\mathcal{S}) \approx \delta p(e|\neg\mathcal{S}) \approx 0$, the slice error given the semantic attribute $\mathcal{S}$ is obtained by separating the two types of observed error probabilities $p(e|\mathcal{C})$, which is possible as long as the denominator is non-zero.[4] An analysis of properties of this equation w.r.t. the denominator allows us to automatically create quality indicators on the validity or invalidity of the obtained corrected slices for attribute $\mathcal{S}$. The full derivation and further details on quality indicators can be found in appendix B.2.

**Weak Slice discovery on Structured ODD Data with SliceLine**: We have now established methods to generate metadata and correct for noise during the metadata generation. With this background, in algorithm 1, we propose three stages for Systematic Weakness Detection (SWD-1,2,3). In SWD-1, using the generated structured metadata and observed errors $p(e|\mathcal{C})$, we employ SliceLine (Sagadeeva & Boehm, 2021)

---

[3]High-quality human labeling typically requires multiple measures to reduce inter-observer variability or epistemic uncertainty in general (e.g. via labeling guides). However, in this work, we consider human labeling as high-quality compared to DNN-based labeling.

[4]For the sake of numeric stability, also denominators which are only approximately zero should be discarded.

to provide a ranked list of top-$k$ worst performing slices based on a scoring function that takes into account the errors and sizes of the slices (see eq. (5) in appendix A.4 for details on how SliceLine works). As we have motivated, observed errors may not always provide a sufficient signal to identify the underlying error (see the top row in fig. 2). Therefore, in SWD-2, using eq. (4) to compensate for noise in the metadata, we provide corrected errors instead of observed errors to SliceLine to provide a second ranked list of top-$k$ worst-performing slices $\mathcal{S}$. However, as it requires extensive human effort to identify certain parameters, i.e., $\delta p(e|\mathcal{S})$, $\delta p(e|\neg\mathcal{S})$ in eq. (4), in particular for combinations of semantics, we make a cheaper approximation only considering the independence assumption part of the equation. This is implemented in `computeCorrectedError()` in algorithm 1. To operationalize this part of the equation, we estimate precision values based on human evaluation of metadata quality on only $n = 60$ samples per attribute (see appendix B.4). The subsequent corrected errors from this independence assumption are used in the SliceLine scoring function. Based on the slice quality indicators discussed above, we are also able to discard invalid slices due to denominator values close to zero. In addition to SWD-1 and SWD-2, we also consider a merge of the resulting slices from SWD-1 and SWD-2, as this might provide a complementary effect. We refer to this merged list as the output of SWD-3. The merge step includes sorting based on the score of the slice from the scoring function, removal of duplicate slices, and filtering of invalid slices. The SliceLine hyperparameters include the level (maximal search depth), i.e. the maximal number of semantic dimensions considered simultaneously, as well as a cut-off for the necessary slice error $\bar{e}|_{\mathcal{S}}$ to consider $\mathcal{S}$ a valid slice.

---

**Algorithm 1:** Systematic Weakness Detector (SWD)

---
**Input:** Metadata $\{\mathcal{C}_{\mathcal{Z}_1}, \mathcal{C}_{\mathcal{Z}_2}, \dots\}$, errors $e$, Precision vectors $\{p_{\mathcal{C}}\}$, SliceLine hyper-parameters
**Output:** Top-K slices $TS$

**1 SWD-1: SliceLine with observed errors $p(e|\mathcal{C}_i)$;**
**2**     $[TS_1] \leftarrow$ SliceLine($\{\mathcal{C}_{\mathcal{Z}_1}, \mathcal{C}_{\mathcal{Z}_2}, \dots\}, e,$ hyperparameters);
**3 SWD-2: SliceLine with corrected errors (approximations to $p(e|\mathcal{S}_i)$);**
**4**     $[TS_2, \text{Quality Indicators}] \leftarrow$
       SliceLine($\{\mathcal{C}_{\mathcal{Z}_1}, \mathcal{C}_{\mathcal{Z}_2}, \dots\},$ **computeCorrectedError**$(e, p_{\mathcal{C}}),$ hyperparameters);
**5 SWD-3: Combined Slices;**
**6**     $[TS] \leftarrow$ Merge($TS_1 \cup TS_2$);
**7 return** TS;

---

## 4 Proof of Concept with Synthetic Data

To demonstrate the efficacy of our algorithm fig. 1, we first present evaluations on a synthetic dataset. This is done to evaluate the impact of noise on the labeling process and to determine the degree to which our algorithm can compensate for it. The synthetic data is a tabular dataset containing columns for nine "real" semantic dimensions for $200\,000$ samples each containing binary attributes. For each of the "real" dimensions (GT), a "predicted" metadata column is included as a proxy for the metadata that would be generated by CLIP in our algorithm (see fig. 1). In addition, one final column contains the binarized **DuT** errors ($e$). The first four dimensions are generated to be imbalanced with only $\sim 5\%$ of the samples belonging to the attribute "1". The other five dimensions are generated such that both attributes have equal distribution. The error column is designed such that weak slices are induced for the specified ground-truth attributes.

We consider three regimes of noise, i.e., different quality of labeling of the simulated annotation process: (i) a regime of "good" quality CLIP labeling, represented with $p_{\mathcal{C}}$ above 80%, (ii) a regime of "medium" quality CLIP labeling, represented with $p_{\mathcal{C}}$ between 40% and 70%, and (iii) a regime of "bad" quality CLIP labeling, represented with $p_{\mathcal{C}}$ between 10% and 40%. For all three regimes, we considered 100 runs of the experiments to account for statistical influence. Further details about the dataset generation can be found in appendix A.2. In fig. 2, in the top row, the error distributions show how labeling quality impacts the spread of error between attributes for each semantic dimension, i.e., the upper and lower ends of the bars are given by the error rates for $\bar{e}|_{\mathcal{S}}, \bar{e}|_{\neg\mathcal{S}}$ and similarly using $\mathcal{C}$ or the corrected errors. In the good labeling quality regime, as expected, observed errors and corrected errors both display the same spread as the GT error. But when labeling quality is medium or bad (where the impact of eq. (4) is stronger), the spread of the observed error is significantly lower than that of the corrected error. In contrast, the corrected error

either has close estimates to the true error or overestimates the true error (GT). From a safety perspective, we argue that overestimating the error within a DNN is better than underestimating it. In the bottom row, we evaluate the results of SWD-1,2,3. This is shown by comparing how well SWD-1,2,3 recover the top-$k$ weak slices in comparison to top-$k$ slices from Oracle, i.e., a situation where we have access to perfect "GT" labeling quality annotation. Precision and recall are calculated for the three data quality regimes w.r.t. the Oracle case by considering the overlap of identified weak slices at an increasing number of top-slices $k$. Note that precision and recall in this figure refer to quality metrics on weak slice discovery and not precision and recall of the CLIP labeling. While, at level 2, the maximum number of slices $k$ is 162 for 9 binarized dimensions[5], we consider only slices fulfilling the cut-off requirement as a weak slice. Of these 162 slices, only $\sim$ 30 are identified as weak slices. Although under good labeling quality, the slices identified by SWD-1,2,3 basically have 100% overlap with the slices from the Oracle, under medium and strong label noise, SWD-3 shows significantly more recall than SWD-1 and marginally over SWD-2. However, this comes with a small loss in precision. In cases of strong noise, SWD-1 only recovers a few slices where the error signal is dominant, which explains the high precision at the cost of low recall. SWD-3, on the other hand, has a reduced precision, but recovers most of the weak slices identified by Oracle. For the rest of this work, we focus primarily on the slices identified by SWD-3.

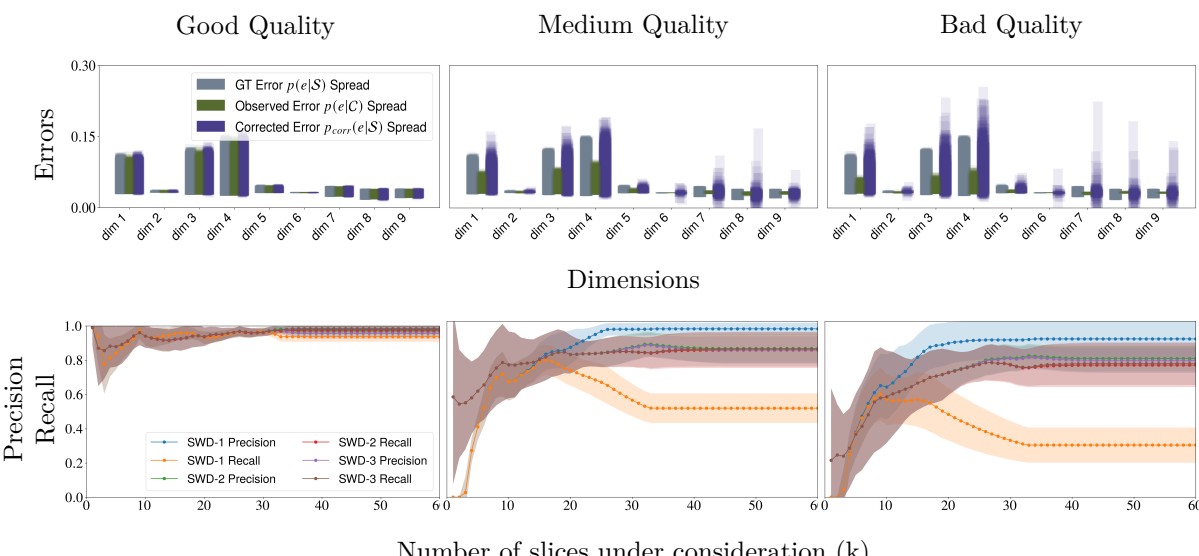

Figure 2: Based on labeling quality, we divide the generated datasets into (i) good quality (left), (ii) medium quality (middle), and (iii) bad quality (right). In three cases, we look at the spread of error in GT ($p(e|\mathcal{S})$), Observed ($p(e|\mathcal{C})$), and Corrected ($p(e|\mathcal{S})$). In the second row, corresponding performance in terms of precision and recall of SWD-1,2,3 are shown. Precision and Recall in this figure are metrics to evaluate weak slice recovery and are not related to labeling quality. The legend for both rows are presented on the figures on left.

# 5   Evaluations of real-world DNNs

In this section, we first present our experimental setup. We then show the evaluation of our systematic weaknesses detection method on a publicly available pre-trained model for the CelebA dataset. Here, the dataset's rich metadata annotation allows us to investigate the influence of noisy metadata annotation. In addition, we compare against SOTA SDM methods to evaluate our claim that adherence to the ODD descriptions is useful to end users (e.g., safety experts, ML developers). Subsequently, we present the insights gained by using our approach on DNNs trained on autonomous driving datasets.

---

[5]$9 \times 2 + \binom{9}{2} \times 2^2$

## 5.1 Experimental Setup

**Datasets and Models:** Five pre-trained models, ViT-B-16 (Dosovitskiy et al., 2021), Faster R-CNN (Ren et al., 2015), SETR PUP (Zheng et al., 2021), PanopticFCN (Li et al., 2021), and YOLOv11m (Jocher & Qiu, 2024) are evaluated using five public datasets (CelebA (Liu et al., 2015), BDD100k (Yu et al., 2020), Cityscapes (Cordts et al., 2016), RailSem19 (Zendel et al., 2019)), and EuroCity Persons dataset (Braun et al., 2019), respectively. We restrict the number of combinations (level) to 2 in this work. However, as presented in appendix B.4, our approach allows correction of errors even at higher levels of combinations. We used the cutoff for the slice error as $1.5\,\bar{e}|_{\mathcal{D}}$ for all experiments except the PanopticFCN model evaluation. In the PanopticFCN evaluation, we utilize the cutoff point for the slice error as $1.0\,\bar{e}|_{\mathcal{D}}$ as the global average error is already quite high. For a detailed experimental setup, see appendix A.1. To foster reproducibility, code, ODDs, and the prompts used for metadata generation with CLIP are provided.

## 5.2 Evaluation of our Systematic Weaknesses Detection Method

**Evaluating a ViT Model on CelebA dataset:** As our first experiment, we evaluated the weaknesses of the ViT-B-16 (Dosovitskiy et al., 2021) model (**DuT**) trained on ImageNet21k (Ridnik et al., 2021). We use the model for the targeted task of identifying the class "person" in the CelebA dataset (Liu et al., 2015) as a real-world proof of concept for our approach. Due to the extensive range of label categories in ImageNet (Deng et al., 2009) and the significant noise in the labeling style, models trained on the full ImageNet dataset or its standard subset ImageNet1k (Russakovsky et al., 2015) can suffer from systematic weaknesses. For example, although the primary foreground object in an image might be a human, in some instances the image can be labeled as belonging to the class "person", while in other similar instances the label might be about more granular classes like "bride" or "guitarist". To fix this issue, Ridnik et al. (2021) proposed 11 hierarchies based on WordNet (Miller, 1995) semantic trees such that classes at higher hierarchy levels are superclasses that subsume classes at lower hierarchy levels. However, despite these efforts, considerable label noise in terms of class overlap still persists. For example, humans holding specific objects might occur at the same hierarchy level as the class "artifact" or "person". Similar problems exist, for example, for hairstyles (see "pompadour" existing at the same level as "person"). For a further analysis, also see the work of (Northcutt et al., 2021).

Earlier works (Beyer et al., 2020; Shankar et al., 2020) have proposed using multi-label evaluation metrics as a way to deal with label noise. However, we consider the simplified task of identifying a dedicated class, "Person", in a dataset with only human faces (CelebA) by focusing on the top-1 class predictions for level-0 of the label hierarchy proposed in ImageNet21k. We obtain an accuracy of 94.48% on the 202 599 images in the CelebA dataset. The softmax of the top-1 prediction, see fig. 3, shows, besides the "person" class, the presence of several other classes, most prominently "artifact" and "pompadour". As this model is commonly used as a pre-trained backbone for various applications, uncovering potential shortcomings might also be beneficial for potential downstream use cases. Furthermore, the CelebA dataset serves as an ideal testing ground for approaches identifying systematic weaknesses due to the availability of the ground-truth metadata attributes. As an ODD for this test case, we propose a simplified subset of these available metadata attributes in analogy to the work of Gannamaneni et al. (2023), for details see appendix A.3. As proposed in our algorithm, we generate metadata using CLIP for the given ODD dimensions. In principle, stronger and more recent models such as GPT-4o (Hurst et al., 2024) could also be used for this task at greater computational (or financial) cost. However, such models might still produce error-prone metadata. A performance comparison is given in appendix E. Subsequently, the generated metadata is combined with the errors of the **DuT**.

**Weak Slice Discovery** Since the CelebA dataset contains annotated metadata for 40 attributes, we have access to noiseless metadata which, when used with SliceLine, can be considered as the "Oracle" approach. In table 1, we present the quantitative comparison of the top-7 slices identified by SWD-3 against corresponding slices in SWD-1 and Oracle. Basically, we list the top-7 slices of SWD-3 and evaluate where these slices would be ranked by SWD-1 and Oracle and what the corresponding statistics would be to highlight the importance of error correction. From the slice descriptions, all the identified weak slices contain some variation of the semantic concept "wearing hat". The discovery of these slices can be seen within the context

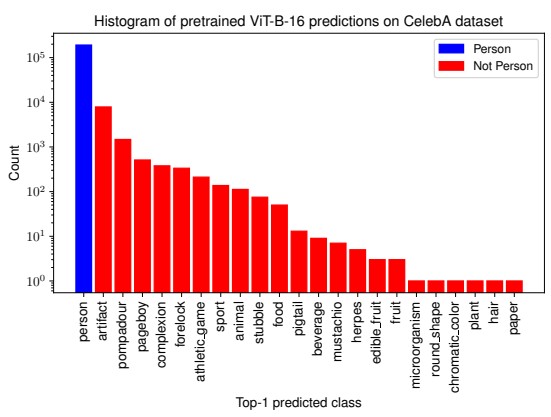

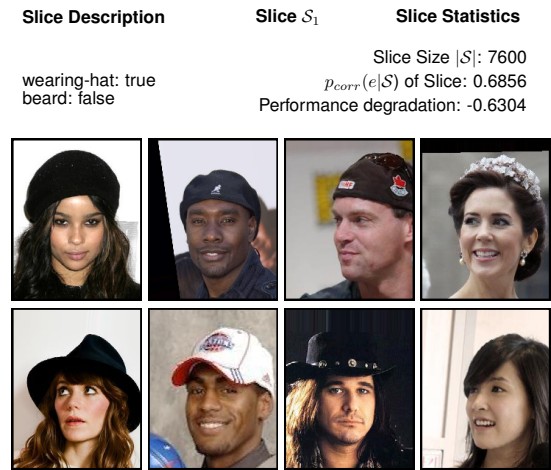

Figure 3: Left: The hierarchy level-0 (Ridnik et al., 2021) predictions of the pre-trained ViT-B-16 model on the full CelebA dataset converted into a binary classification problem. While a majority of the predictions are correct, there is a non-trivial subset of images with systematic errors due to label overlap issues. Right: Top-1 weak slice, identified by SWD-3, of a ViT-B-16 classification model trained on ImageNet21k and evaluated on the full CelebA dataset. The statistics provide a quantitative evaluation of the entire slice. For qualitative evaluation, we provide some sample images from the slice.

of the frequent misclassification of images as class "artifact" by the **DuT** (see fig. 3). In these cases, the model likely focuses on the hats as the foreground object and predicts the class "artifact". To evaluate the quality of the identified slices, we utilize the errors of the slices, i.e., $p_{corr}(e|\mathcal{S})$, $p(e|\mathcal{C})$, and $p(e|\mathcal{S})$ defined in section 3. We observe, based on the rank column, that the top-6 Oracle slices are captured in top-7 SWD-3 slices. Notably, while the observed error $p(e|\mathcal{C})$ of SWD-1 underestimates the true error $p(e|\mathcal{S})$ of Oracle, SWD-3 effectively corrects this in $p_{corr}(e|\mathcal{S})$. For instance, in the third row, which corresponds to the top-ranked weak slice identified by the Oracle, the difference between the Oracle slice error and the SWD-1 slice error is 0.3, while between SWD-3 and Oracle it is only 0.07. A thorough evaluation of our approach on the top-60 slices shown in fig. 5 in appendix C.1 reveals that SWD-3 obtains 100% recall of weak slices at the cost of a reduction in precision. Note that precision and recall in this figure refer to quality metrics on weak slice discovery and not precision and recall of the CLIP labeling. From a safety perspective, given the noisy labeling, high recall (detection of all weak slices) at the cost of some reduction in precision can be considered acceptable. Interestingly, the top-1 slice of SWD-1 (not shown in the table) refers to slice description "wearing hat: true" and "pale-skin: true". Gannamaneni et al. (2023) discussed the limitations of CLIP in separating the latter dimension and corresponding low performance. This high level of noise in the generated metadata leads to SWD-1 identifying "pale-skin" as a top-1 slice while SWD-3 effectively corrects for this by discarding the wrongly detected slice as "invalid" using the quality indicators (see algorithm 1) and hence does not identify this dimension in top-7. For a qualitative evaluation of SWD-3, the top-1 slice with sample images from the slice are available in fig. 3 (see fig. 6 in appendix C.2 for a qualitative evaluation of the top-5 slices).

**Comparison to SOTA SDM Methods:** In addition to the evaluation of SWD-3, we compare three SOTA methods DOMINO (Eyuboglu et al., 2022), Spotlight (d'Eon et al., 2022), and SVM FD (Jain et al., 2023) against Oracle. Similarly to our work, DOMINO and SVM FD use CLIP (ViT L/14) in their workflows. However, they encode the images in the CLIP embedding space and then search for weak slices without explicitly enforcing any semantic concepts. To describe the slices, both approaches perform an additional step, where the identified slices are explained using text from large language models. In contrast, Spotlight directly uses the embedding space of the **DuT** to cluster weak slices and provides no descriptions of the identified slices. The former methods follow a broader trend (like us) of using foundation models like CLIP in testing smaller models. However, they do not thoroughly address limitations in CLIP's capabilities and the limitations of their approaches w.r.t. actionability when the slice descriptions are not very meaningful.

| Slice | Slice Description | SWD-3 | | | SWD-1 | | | Oracle | | |
|---|---|---|---|---|---|---|---|---|---|---|
| | | $\text{rank}(S)$ | $|\mathcal{S}|_{\text{corr}}$ | $p_{\text{corr}}(e|\mathcal{S})$ | $\text{rank}(S)$ | $|\mathcal{S}|$ | $p(e|\mathcal{C})$ | $\text{rank}(S)$ | $|\mathcal{S}|$ | $p(e|\mathcal{S})$ |
| $\mathcal{S}_1$ | **Wearing-Hat**: True **Beard**: False | 1 | 7600 | 0.69 | 6 | 12152 | 0.33 | 2 | 6267 | 0.51 |
| $\mathcal{S}_2$ | **Wearing-Hat**: True **Smiling**: False | 2 | 5132 | 0.60 | 3 | 8573 | 0.36 | 9 | 6476 | 0.45 |
| $\mathcal{S}_3$ | **Wearing-Hat**: True **Gender**: Female | 3 | 4435 | 0.61 | 2 | 7393 | 0.38 | 1 | 2947 | 0.69 |
| $\mathcal{S}_4$ | **Wearing-Hat**: True **Age**: Young | 4 | 7974 | 0.54 | 4 | 12758 | 0.34 | 3 | 6937 | 0.50 |
| $\mathcal{S}_5$ | **Wearing-Hat**: True **Eyeglasses**: False | 5 | 8606 | 0.54 | 5 | 12594 | 0.33 | 6 | 8417 | 0.45 |
| $\mathcal{S}_6$ | **Wearing-Hat**: True **Goatee**: False | 6 | 8845 | 0.53 | 7 | 11453 | 0.33 | 4 | 8284 | 0.46 |
| $\mathcal{S}_7$ | **Wearing-Hat**: True **Bald**: False | 7 | 9676 | 0.51 | 8 | 15501 | 0.32 | 5 | 9795 | 0.44 |

Table 1: Evaluation of top-7 slices of SWD-3 (see algorithm 1) by comparing its statistics with corresponding slice statistics of SWD-1 and Oracle. The rank column indicates the slice ranking in each approach. The size of $\mathcal{D}$ for this experiment is $202\,599$ images.

Our approach tackles both these limitations as we address noise in CLIP labeling and also correctness of descriptions. However, our approach also has a limitation as it can only identify weaknesses w.r.t. dimensions in $\mathcal{Z}$ while the other methods could identify more novel weaknesses. However, this advantage of SOTA methods, as will be shown below, can only be realized if the description or coherence of a slice is understandable and actionable to the end-user. To assess the actionability of the SOTA methods, we consider (i) slice descriptions based on the methods themselves, (ii) slice coherence based on human inspection, and (iii) slice coherence based on overlap with top-5 slices of Oracle.

Qualitative results and slice descriptions are provided for the three methods in appendix C.2. We identified that DOMINO descriptions can be very generic and not helpful in identifying the unique attributes of a slice. This problem was also discussed in other works (Jain et al., 2023; Gao et al., 2023). For Spotlight, descriptions are not available as part of the method. In contrast, in SVM FD, the slice description is targeted and covers one dimension of the weak slice identified by Oracle, namely, "wearing hat". However, as shown earlier, the weaknesses identified from Oracle stem from the combination of semantics. Therefore, slice descriptions from the SOTA methods are not enough for actionability. Second, to further evaluate the coherence of the slices, we manually inspect a sample of the images from a slice to identify the semantics. Such an approach is necessary for all methods that do not provide slice descriptions. For such manual inspection to identify the coherence of the slice, we consider samples from the slice and samples from the remaining data (last column) as a form of control group. For top-1 slices of all three approaches, it is hard to determine what uniquely constitutes the top-1 slice when considering the combination of semantics. Furthermore, such an exercise is time intensive and might potentially uncover spurious patterns.

Finally, to evaluate the coherence of the slice based on overlap with Oracle slices, we present in table 2 the top-1 slice identified by each method, their corresponding statistics and the overlap (Jaccard Similarity Coefficient) of the top-1 slice with the top-5 Oracle slices. From the slice statistics, it can be observed that the methods recover slices with significant performance degradation and observed error $p(e|\mathcal{C})$. However, the overlap of the top-1 slices with top-5 of Oracle is quite low. This indicates that the methods might be uncovering weaknesses w.r.t. dimensions not present in the ODD. However, without useful descriptions,

| Method | Slice Statistics | | | Slice Coherence with Attributes | | | | | Overlap with FNs |
|---|---|---|---|---|---|---|---|---|---|
| | Perf. degr. | Size | | Overlap with Oracle top-5 slices | | | | | |
| | $\bar{e}\|_{\mathcal{D}}$ - $p(e\|\mathcal{C})$ | $\|\mathcal{S}_1\|$ | $p(e\|\mathcal{C})$ | $J(\mathcal{S}_1, \mathcal{S}_1^O)$ | $J(\mathcal{S}_1, \mathcal{S}_2^O)$ | $J(\mathcal{S}_1, \mathcal{S}_3^O)$ | $J(\mathcal{S}_1, \mathcal{S}_4^O)$ | $J(\mathcal{S}_1, \mathcal{S}_5^O)$ | $\frac{\|\mathcal{S}_1 \cap \mathcal{S}_{FN}\|}{\|\mathcal{S}_{FN}\|}$ |
| DOMINO | -0.5629 | 11726 | 0.6181 | 0.13 | 0.19 | 0.20 | 0.21 | 0.24 | 0.64 |
| Spotlight | -0.8622 | 4050 | 0.9179 | 0.32 | 0.32 | 0.32 | 0.31 | 0.31 | 0.33 |
| SVM FD | -0.3844 | 2642 | 0.4295 | 0.11 | 0.15 | 0.16 | 0.16 | 0.16 | 0.24 |

Table 2: Comparison of three metadata-free SOTA methods with top-5 slices of Oracle. $J(\mathcal{S}_1, \mathcal{S}_x^O)$ indicates the Jaccard similarity coefficient between the two slices. appendix C.2 contains samples from each slice of the SOTA methods along with slice descriptions and statistics. For overlap with the oracle slice, higher values are better. For the overlap with the FNs, low values indicate that a slice does not contain "significant" weaknesses or is highly specific, while high values indicate that potentially all weaknesses of the **DuT** are in one slice and it might, therefore, be too generic. This implies that in general one would expect or desire medium overlap ranges.

the actionability of these slices is low. Furthermore, we evaluate the overlap of the top-1 slice with a slice that is purely made up of FNs of the **DuT**. High values in this column might indicate that priority is given to identifying FNs rather than semantic coherence, as it is unlikely that all weaknesses of a DNN can be explained by one semantic concept. Therefore, grouping all FNs into one slice would be counterproductive. As DOMINO captures 64% of all false negatives in its top-1 slice, it is unlikely that such a slice is actionable. In contrast, Spotlight and SVM FD capture fewer FNs in the top-1 slice. Therefore, they might be capturing some form of combination of semantics. Based on these evaluations, we conclude that the SOTA methods, when integrated with improved slice description techniques, could complement our approach. However, in their current form, our approach offers greater actionability due to its inherent slice descriptions.

## 5.3 Insights on SOTA Pedestrian Detection Models

Having shown the benefits of our proposed method, we evaluate a more safety-relevant task of pedestrian detection using models trained in real-world autonomous driving (AD) datasets to identify their systematic weaknesses when predicting the class "pedestrian". For this, we require pedestrian level performances (intersection-over-union (IoU)) and metadata. To avoid noisy labeling in our metadata generation step, we perform some additional steps which were not required for the previous experiment. First, we cropped all pedestrians from the images and considered these crops as $\mathcal{D}$. This is done to focus the CLIP model only on pedestrians during metadata generation.[6]

Second, we calculate the pixel area of the pedestrians based on the ground truth bounding box area and use this to filter $\mathcal{D}$ by removing pedestrians that occupy small pixel areas ("smaller" pedestrians). Such filtering is necessary as: (i) Due to low resolution and high pixelation of "smaller" sized pedestrians, i.e., there is a high aleatoric uncertainty regarding the correct labels affecting both CLIP and human labelers in understanding the image content (e.g., to determine gender, age, etc.). (ii) "Smaller" pedestrians are more likely to be farther from the ego-vehicle [7] and, therefore, might be considered less safety-relevant (in terms of vehicle braking time). (iii) As the small size can be strongly correlated to performance (due to distance (Gannamaneni et al., 2021; Lyssenko et al., 2021) or occlusion), this signal can strongly dominate the search for systematic weaknesses by SliceLine, thus not providing any novel insights in terms of systematic weaknesses. For this reason, we remove the low-resolution "smaller" pedestrians to improve the quality of metadata generation and gain further novel insights about model failures w.r.t. more safety-relevant pedestrians.

---

[6]To avoid that the aspect ratio of pedestrians is changed by the CLIP pre-processing, we use padding to obtain square crops.
[7]Unless if small size is due to occlusion. For BDD100k dataset, where occlusion is available as annotation, we show impact of occlusion as well

The metadata generation using CLIP is performed using ODDs more suitable for automotive context (see appendix A.3). We also perform a manual evaluation of a subset of images ($n = 60$) for each attribute in each dataset to evaluate the quality of the generated metadata by estimating the precision $p_\mathcal{C}$ and recall $r_\mathcal{C}$ (as discussed in section 3) and show the results in table 3.

| Sem. dim. | Attri. | Estimated Precision $p_\mathcal{C}$ | | | Estimated Recall $r_\mathcal{C}$ | | |
|---|---|---|---|---|---|---|---|
| | | BDD100k | Cityscapes | RailSem19 | BDD100k | Cityscapes | RailSem19 |
| Age | Adult | $0.95 \pm 0.03$ | $0.99 \pm 0.02$ | $0.97 \pm 0.02$ | $0.76 \pm 0.03$ | $0.70 \pm 0.02$ | $0.55 \pm 0.02$ |
| | Young | $0.69 \pm 0.06$ | $0.56 \pm 0.06$ | $0.42 \pm 0.06$ | $0.93 \pm 0.06$ | $0.97 \pm 0.06$ | $0.94 \pm 0.06$ |
| Gender | Female | $0.84 \pm 0.05$ | $0.97 \pm 0.02$ | $0.85 \pm 0.04$ | $0.90 \pm 0.05$ | $0.95 \pm 0.02$ | $0.87 \pm 0.04$ |
| | Male | $0.94 \pm 0.03$ | $0.97 \pm 0.02$ | $0.94 \pm 0.03$ | $0.88 \pm 0.03$ | $0.97 \pm 0.02$ | $0.92 \pm 0.03$ |
| Cloth.-color | Bright-color | $0.81 \pm 0.05$ | $0.85 \pm 0.04$ | $0.79 \pm 0.05$ | $0.30 \pm 0.05$ | $0.23 \pm 0.04$ | $0.66 \pm 0.05$ |
| | Dark-color | $0.76 \pm 0.05$ | $0.65 \pm 0.06$ | $0.82 \pm 0.05$ | $0.96 \pm 0.05$ | $0.97 \pm 0.06$ | $0.89 \pm 0.05$ |
| Skin-color | Dark | $0.82 \pm 0.05$ | $0.55 \pm 0.06$ | $0.56 \pm 0.06$ | $0.92 \pm 0.05$ | $0.71 \pm 0.06$ | $0.76 \pm 0.06$ |
| | White | $0.99 \pm 0.02$ | $0.95 \pm 0.03$ | $0.89 \pm 0.04$ | $0.96 \pm 0.02$ | $0.91 \pm 0.03$ | $0.75 \pm 0.04$ |
| Blurry | True | $0.71 \pm 0.06$ | $0.63 \pm 0.06$ | $0.87 \pm 0.04$ | $0.42 \pm 0.06$ | $0.87 \pm 0.06$ | $0.64 \pm 0.04$ |
| | False | $0.48 \pm 0.06$ | $0.95 \pm 0.03$ | $0.84 \pm 0.05$ | $0.74 \pm 0.06$ | $0.82 \pm 0.03$ | $0.95 \pm 0.05$ |
| Constru.-Worker | False | - | - | $0.97 \pm 0.02$ | - | - | $0.98 \pm 0.02$ |
| | True | - | - | $0.65 \pm 0.06$ | - | - | $0.55 \pm 0.06$ |

Table 3: The estimated precision and recall using our proposed approach for evaluating the quality of the generated metadata. Here, we provide the mean and $\sigma/2$, for $n$ of 60, of the estimated precision and recall. Certain dimensions like occlusion are available as part of the datasets themselves. We do not perform human-evaluation for these dimensions but these are considered in the weak slice search.

In these experiments, using SWD-3, we evaluate the weaknesses of an object detection model (Faster R-CNN), a segmentation model (SeTR PUP), and a panoptic segmentation model (Panoptic-FCN). The models are evaluated on their respective datasets, i.e., BDD100k, Cityscapes, and RailSem19. Samples of image crops of the identified top-1 weak slice for each experiment are shown in fig. 4 (see figs. 10 to 12 in appendix C.2 for top-5 weak slices). In table 4, we present the largest and worst performing slice of the top-5 to provide insights about the three models. In all three experiments, the performance degradation of the identified slices is significant. "Occlusion", skin-color and clothing-color are reoccurring slice descriptions for the first two models, which are tested on datasets that contain images with many nighttime scenes (BDD100k) or relatively high gray-toned scenes (Cityscapes). In contrast, the third model, which contains relatively brighter scenes, has a significant weakness for the dimension "age". The estimated precision $p_\mathcal{C}$ and recall $r_\mathcal{C}$ in table 3 were provided as input to algorithm 1 to obtain these slices and to determine the quality of the identified weaknesses. Therefore, in contrast to SOTA SDMs, our approach identifies human-understandable safety-relevant systematic weaknesses in DNNs used for real-world applications. To evaluate scalability, we also investigate systematic weaknesses in the YOLOv11m (Jocher & Qiu, 2024) model by treating it as the **DuT** and evaluating it on the comparatively larger EuroCity Persons (Braun et al., 2019) dataset. The results are presented in appendix D.

# 6 Conclusion

In this work, we present an algorithm for our Systematic Weakness Detector (SWD) to analyze the systematic weaknesses of DNNs that perform classification, object detection, and semantic segmentation tasks on image data. In the first step, we overcome the problem of missing metadata by generating metadata with a foundation model. Subsequently, in the second step, we perform slice discovery using the structured metadata, which includes the per-object performance of the **DuT** and the previously collected per-object metadata. Using our algorithm, we transform the slice discovery of unstructured image data into an (approximate) slice discovery problem on structured data. In addition, we study the impact of noisy labeling in

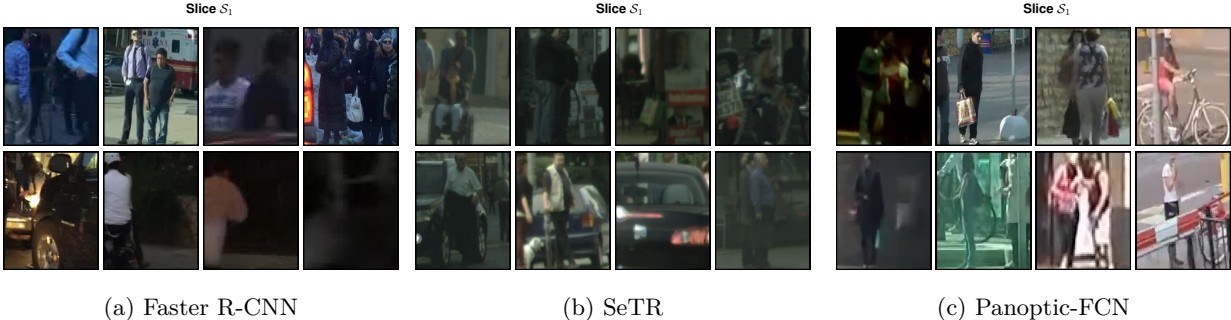

(a) Faster R-CNN        (b) SeTR        (c) Panoptic-FCN

Figure 4: Left: Samples from top-1 weak slice of a Faster R-CNN object detector trained and evaluated on BDD100k dataset. Middle: Samples from top-1 weak slice of SeTR model trained and evaluated on Cityscapes dataset. Right: Samples from top-1 weak slice of a Panoptic-FCN model trained and evaluated on RailSem19 dataset.

a Bayesian framework and operationalize it by integrating error correction and slice validity based on quality indicators into our approach.

**Main Results:** In the ablation experiments, we show that our SWD detects the same weak slices as would be identified in hypothetical cases where we have access to perfect metadata. The primary advantage of our algorithm, in comparison to SOTA methods, is that the identified weak slices are aligned with human-understandable semantic concepts that can be derived from a description of the ODD. As upcoming safety and trustworthy AI specifications increasingly mandate evidence to support safety argumentation with respect to specific ODDs, the results of our approach can serve as a direct contribution to these requirements. In addition, the identification of human-understandable weak slices enables ML developers to take mitigation actions, such as a targeted acquisition or generation of data, addressing the weaker slices and, thus, facilitating effective re-training with a limited acquisition budget. Furthermore, we show that our approach has clear advantages over several metadata-free SOTA methods by giving more actionable results, and we demonstrate the applicability of our approach by identifying systematic weaknesses in multiple AD datasets. For this, we also provide a quantitative evaluation of the quality of the generated metadata.

**On metadata generation and incompleteness of ODDs:** A minimum metadata labeling quality is required for the discovered slices to be meaningful. In addition to our proposed metadata quality estimation, future works could therefore focus on improving metadata quality by human correction of a subset of generated metadata, fine-tuning of CLIP (Eyuboglu et al., 2022), and metadata acquisition from other sources (e.g., depth sensor for distance). Secondly, all approaches based on ODD definitions, like ours or PromptAttack (Metzen et al., 2023), would suffer from the lack of completeness of the semantic concepts

| Model & Dataset | Largest Slice (in top-5) | | | Worst Performing Slice (in top-5) | | |
|---|---|---|---|---|---|---|
| | $\frac{|\mathcal{S}|}{|\mathcal{D}|}\%$ | $p_{\text{corr}}(e|\mathcal{S})$ | Perf. Degr. | $\frac{|\mathcal{S}|}{|\mathcal{D}|}\%$ | $p_{\text{corr}}(e|\mathcal{S})$ | Perf. Degr. |
| Faster R-CNN BDD100k | 34.44% | 0.1263 | -0.0693 | 14.22% | 0.2206 | -0.1636 |
| SeTR Cityscapes | 13.34% | 0.0594 | -0.0446 | 9.24% | 0.1046 | -0.0897 |
| Panoptic-FCN RailSem19 | 25.49% | 0.8663 | -0.222 | 8.13% | 1.0 | -0.4602 |

Table 4: Quantitative analysis of three pre-trained autonomous driving models (results are only for SWD-3). From the top-5 weak slices, we show the largest slice and the weakest performing slice. Please refer to the appendix C.2 for the top-5 slices.

in the ODD. A potential solution could be in the direction of Gannamaneni et al. (2024) by performing a root-cause analysis of found weaknesses. Such approaches could address potential issues between correlation and causation for found small slices.

**On generation of ODDs:** ODDs are a critical component in the functioning of our algorithm, necessitating careful consideration of their availability. In safety-critical domains such as automotive, medical, and aerospace applications, we posit that domain and safety experts possess the requisite expertise to develop ODDs. In contrast, for use cases lacking predefined ODDs, the use of large language models (LLMs) to support ODD generation presents a promising direction, akin to the tag generation approach proposed in (Chen et al., 2025). Nevertheless, we emphasize that human oversight remains essential, both in the final approval of the generated ODDs and in their subsequent refinement and extension.

**On considering foundation models as DuT:** In this work, we evaluate state-of-the-art (SOTA) architectures for classification, segmentation, and object detection tasks within the framework of **DuT**, leveraging a more "powerful" foundation model (CLIP) as a metadata generation function. However, the proposed algorithm can also be employed to identify systematic weaknesses and biases inherent in foundation models themselves, by treating the foundation model as the **DuT**. For instance, if the foundation model is intended for tumor detection in X-ray scans, then suitable choices of $\mathcal{Z}$ and $\mathcal{D}$ must be determined to enable the application of the algorithm.

**On false positives:** This work focuses on false negatives for two main reasons: first, in safety-critical tasks, mitigating false negatives is generally more crucial than addressing false positives; second, the ODDs we develop, based on input from domain experts, are specifically centered around pedestrian-related semantics and their attributes. Addressing false positives would require expanding the ODDs to encompass all non-pedestrian classes that could potentially be misclassified. This challenge is both an open-set problem and highly dependent on the specific **DuT** under consideration. If false positives are to be incorporated within our algorithm, an initial exploratory analysis would be helpful to understand their distribution. For example, in the RailSem19 experiment, we observed that a large portion of false positives involved railway signs being misclassified as pedestrians. Based on such analyses, new ODDs (e.g., based on object type, visual features like color, etc.) could be developed for these frequently misclassified objects, after which our algorithm could be applied. The exploratory analysis could be based on concept-focused clustering approaches, compare, e.g., Haedecke et al. (2024), or even use the outcome of slice-and-tag approaches. Nevertheless, even when focusing only on false negatives, our approach provides meaningful insights into, often more critical, missed detections in terms of human-understandable and, thereby, actionable weak slices. We believe that such results can contribute to the development of trustworthy AI models.

# 7 Acknowledgments

This work has been funded by the European Union and the German Federal Ministry for Economic Affairs and Climate Action as part of the safe.trAIn project.

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
