# OpenReview forum: "Detecting Systematic Weaknesses in Vision Models along Predefined Human-Understandable Dimensions"
_TMLR — Accepted by TMLR_

### Review · Reviewer_56t6 · 2025-04-17

**Summary Of Contributions:**

This paper proposes to use a combination of a foundation model (CLIP) for zero-shot classification to generate semantic meta-data with a slice search method (SliceLine) for searching slices of data, for which a trained network fails in predicting the correct target.  The combination of a foundation model and the Slice search method allows for finding slices which are in line with human understanding. In addition, this paper includes a step to handle the uncertainty in prediction of the foundation model in the classification tasks. Several extensive experiments on synthetic data as well as image data for classification and detection show improvements against other methods searching for slices using structured data.

**Audience:**

Yes

**Broader Impact Concerns:**

As mentioned in the weaknesses two important discussion points would are needed:
- A deeper discussion on how to handle false positive systematic error is missing.
- How can biases from pretrained models and foundation models be identified and potentially overcome?

**Claims And Evidence:**

Yes

**Requested Changes:**

- It would be helpful to provide a clearer explanation of why converting to structured data is superior to using approaches with unstructured data.
- A rewrite of the sections mentioned (in section 3) is needed for better clarity.
- A clearer description of the metadata is would be helpful.
- How could the approach be adapted if no ODDs are provided? Could these be generated using an LLM?
- A deeper  discussion on how to handle false positive systematic error is missing.
- How can biases from pretrained models and foundation models be identified and potentially overcome?

**Strengths And Weaknesses:**

**Strengths**:

- This paper provides a strong motivation for identifying samples where the network’s performance falls below the average test performance. By doing so, one can potentially improve the network’s stability and robustness during training.

- The approach allows for pinpointing slices with semantics aligned with human understanding concepts, revealing the areas where the network’s weaknesses lie, offering opportunities for improvement. This approach is particularly valuable for developing better networks or emphasising the slice data to improve the performance, especially in safety-critical domains like autonomous driving.

- Extensive experiments demonstrate the effectiveness of this approach in identifying slices where the network shows weaknesses.

**Weaknesses**:
- A clearer explanation of the generated meta-data would be helpful. It’s not entirely clear to me how CLIP is used and how they differ from the ODD data.
- This paper misses comparisons to other methods that directly use images as unstructured data, such as PromptAttack.  A comparison with these methods could potentially strengthen the need for the step of converting unstructured image data into structured data.
- It would be better to make it clear earlier that $\mathcal{Z}$ are the ODD information.
- This approach is based on the need for ODD. However, how can it handle the case where no ODD are available?
- The last sentence on page 4 is not clear to me: „As such, it therefore provides the coherence of the slices discussed above.“
-  How does this approach address the possibility of systematic false positives?
 - How can the biases from pretrained models and foundation models be identified and potentially overcome? A discussion of this potential harm would be needed.

*Clarity*:

- In my opinion, a rewrite of the „metadata generation“ section is needed. It’s unclear and feels like an overstatement of the sophistication of simply using CLIP for zero-shot classification. The explanation of what the metadata are is still not clear.
- The section of the Bayesian framework contains unclear and loose statements. For example, in my understanding, $p(e|S)$ represents the expected error given the samples in slice $S$, based on the ground truth target from the dataset $\mathcal{D}$. On the other hand, $p(e|C)$ represents the expected error given the samples from slice $S$, but calculated using the prediction of CLIP instead of the DNN. If this is correct, then this section switches between the usage of „expected error“,„true error,“ and the correct usage of probabilities and actual errors. A rewrite of this section is needed to provide a proper mathematical description and usage of the individual notations.
- Page 8, Section 5.2: The first two paragraphs could be shortened and combined, since the first paragraph feels rather distracting from what this experimental setup is actually doing here.

*Questions*:
- Section 5.1: What is the cutoff for the slice error?
- Section 5.1 Are the ODD the target attributes from the CelebA dataset?
-  Table 1: Was the complete CelebA dataset used as test-data for DuT? In any case, an inclusion of the portion of the slice to the actual data $\mathcal{D}$ would be helpful.
-  Table 2: How large is the overlap between the slices found with SWD-3 and the oracle?
- How bad/weak are the actual predictions of the slice data?

---

> ### Author Response · Authors · 2025-04-30
> **Response to reviewer**
>
> We thank the reviewer for their time and detailed feedback and for highlighting the strengths of our work. We address the mentioned weaknesses and requested changes below:
>
> ## Weaknesses (W)
>
> ### W1:	A clearer explanation of the generated meta-data would be helpful. It’s not entirely clear to me how CLIP is used and how they differ from the ODD data.
>
> We have updated the metadata generation section to clarify the usage of CLIP with an example (see changes in blue in Section 3, subsection on metadata generation). In short, we leverage CLIP's zero-shot classification capabilities to classify, for each image $s$, the attributes present across all relevant semantic dimensions. Concerning the relation between metadata and ODD, we deliberately define the ODD description, denoted $\mathcal Z$, to allow a direct correspondence between its elements, i.e., the semantic dimensions and their respective classification tasks. In the text we therefore use ODD and $\mathcal Z$ interchangeably, although, as pointed out in the newly added footnote 2, an ODD might be broader (see the example there).
> Looking at the datasets, for CelebA, rich annotation is available as GT and we use it as such to demonstrate the effectiveness of our approach. However, for many other datasets, e.g., the AD datasets used in our work, such information is not readily available, therefore making our metadata generation approach necessary to perform the proposed systematic weakness tests.
>
> ### W2:	This paper misses comparisons to other methods that directly use images as unstructured data, such as PromptAttack. A comparison with these methods could potentially strengthen the need for the step of converting unstructured image data into structured data.
>
> We have updated Section 3 to provide a simplified comparison with existing methods (now also including PromptAttack) using our notation (as also requested by reviewer wGv7). (see changes in blue in Section 3, subsection on ``notation''). The main takeaway is that slice-and-tag approaches like DOMINO, Spotlight and SVM-FD do not necessarily produce coherent slices with respect to any specification $\mathcal Z$ as they do not enforce it. The impact of this can be observed in SOTA SDM evaluations in table 2 and appendix C.2. Furthermore, they do not address noise in the used embedding models like CLIP. Tag-and-slice approaches like Ours, PromptAttack and Hibug2 have an advantage, as these approaches specially look for weaknesses aligned with $\mathcal Z$. However, in PromptAttack, diffusion-based generative models are used to create novel data points $s'$ that do not exist in the dataset $\mathcal D$. While this offers a novel way to evaluate weaknesses in the DNN-under-test (DuT), issues related to synth2real gap or diffusion model biases must be taken into account. Our approach is built to identify weaknesses on the given $\mathcal D$ and does not require to build a new set $\mathcal D'$.
>
> ### W3:	It would be better to make it clear earlier that Z are the ODD information.
>
> We have updated the notations in the second paragraph of section 3 to address this.
>
> ### W4: This approach is based on the need for ODD. However, how can it handle the case where no ODD are available?
>
> Our motivation for this work primarily stems from the automotive domain [1, 2], where the ODDs are defined by safety experts with domain knowledge. We believe that for safety-critical systems, e.g., in medical or automotive domains, experts in the field have prior experience in defining the scope of systems and can formalize this experience in ODD definitions. While ODDs could also be generated by LLMs, as shown in the concurrently released HiBug2 [3] with a GPT model, a human evaluation of the ODD would still be necessary. We have updated the conclusion to address this (see changes in blue in "On generation of ODDs".
>
> ### W5: The last sentence on page 4 is not clear to me: "As such, it therefore provides the coherence of the slices discussed above."
>
> In response to W1 above, we have updated this entire paragraph.

---

> ### Author Response · Authors · 2025-04-30
> **Continuation of response to reviewer (1)**
>
> ### W6: How does this approach address the possibility of systematic false positives?
>
> We have revised the second paragraph of the Conclusion section to include a more detailed discussion of the limitations related to ODDs, metadata generation, and false positives (see changes in blue in the Conclusion in "On false positives:").
> Regarding the limitations with respect to false positives (FPs), our work primarily focuses on systematic weaknesses related to false negatives (FNs), as these are considered more safety-critical in the pedestrian detection task. Specifically, failing to detect a pedestrian on the road presents a significantly higher risk than incorrectly identifying a non-pedestrian object, such as a streetlight, as a pedestrian. Therefore, we focus on metadata sets $\mathcal Z$ that specially contain dimensions related to pedestrians themselves. However, as rightly pointed out by the reviewer, weaknesses related to FPs are also relevant from the perspective of model developers and auditors. However, given an autonomous driving scene, a wide array of objects or classes can be misclassified as pedestrians. Therefore, defining $\mathcal Z$ for false positives is non-trivial. We can potentially refine this to a manageable task by first performing an explorative analysis of the FPs of a DNN-under-test (DuT) and then building $\mathcal Z$ for the commonly occurring FPs. For example, in our RailSem19 experiment, we noticed that the DNN-under-test (DuT) has FPs commonly on railway crossing sign posts. Having this knowledge, we would build a dedicated $\mathcal Z_\text{FP}$ that contains dimensions like ``type-of-railway-sign'', ``color-of-railway-sign'', etc.
>
> ### W7: How can the biases from pretrained models and foundation models be identified and potentially overcome? A discussion of this potential harm would be needed.
>
> If the concern pertains to biases in foundation models during metadata generation using $\mathcal{G}$ (e.g., CLIP or GPT), our Bayesian error correction mechanism is explicitly designed to handle imperfect labels resulting from limitations in $\mathcal{G}$. This includes potential biases or systematic weaknesses inherent in CLIP or GPT.
> Two scenarios are possible: (i) The biases of the DuT and $\mathcal{G}$ (e.g., CLIP) are correlated. (ii) The biases of the DuT and $\mathcal{G}$ are uncorrelated.
> We refer to Appendix B.2 for a detailed discussion on the impact of such correlations.
>
> However, if the concern relates to the evaluation of foundation models themselves for biases and systematic weaknesses, our algorithm can be directly applied by treating the foundation model as the DNN-under-test (DuT). In this case, an appropriate task must be selected for evaluation, and the corresponding ODD must be defined.
> For example, to test a GPT model for systematic weaknesses in the task of tumor detection in X-rays, one would need to select an appropriate dataset $\mathcal{D}$, define a suitable ODD $\mathcal{Z}$, and generate the necessary metadata using a metadata generation function $\mathcal{G}$. In this way, our algorithm can be employed to identify whether the GPT model exhibits systematic weaknesses along certain semantic dimensions, such as the tumor's location. See changes in blue in the Conclusion section in "On considering foundation models as DuT".

---

> ### Author Response · Authors · 2025-04-30
> **Continuation of response to reviewer (2)**
>
> ## Clarity (CL)
>
> ### CL1: In my opinion, a rewrite of the "metadata generation" section is needed. It’s unclear and feels like an overstatement of the sophistication of simply using CLIP for zero-shot classification. The explanation of what the metadata are is still not clear.
>
> Following W1, we have updated the metadata generation subsection with an example. Regarding the sophistication of CLIP, we believe that metadata generation for images can be seen as a closed set captioning. Any model capable of zero-shot classification ranging from CLIP to the latest GPT model should be capable of solving this task at various levels of performance. As also presented to reviewer CMrK, our motivation for using CLIP is its lightweight nature and prevalent usage in existing Slice discovery methods like DOMINO [4] and SVM-FD [5]. HiBug2 [3] which came out concurrently to our work uses a GPT model to perform the same task. While such a model is expected to perform better due to their sheer size and generality, we do not expect such solutions to scale in terms of cost. However, we will perform and update the paper with a comparison of CLIP and GPT on the CelebA dataset in a similar experiment setup to Gannamaneni et al. [6]. Details will be given in the Appendix. We will provide the update in the coming days.
>
> Regarding the concretely used metadata dimensions, for celebA the attributes can be found in Table 5, while for the AD experiments they are given in Table 6 (both in Section A.3).
>
> ### CL2: The section of the Bayesian framework contains unclear and loose statements. For example, in my understanding p(e|S), represents the expected error given the samples in slice S, based on the ground truth target from the dataset D. On the other hand, p(e|C) represents the expected error given the samples from slice S, but calculated using the prediction of CLIP instead of the DNN. If this is correct, then this section switches between the usage of ``expected error'',``true error'', and the correct usage of probabilities and actual errors. A rewrite of this section is needed to provide a proper mathematical description and usage of the individual notations.
>
> We have corrected one instance of "excepted error" to "true error". Now, p(e|S) consistently refers to the true error, that is the error of the DuT, when tested on the slice S of the dataset. Regarding p(e|C), we would like to clarify the statement "prediction of CLIP instead of the DNN" by the reviewer. p(e|C) refers to the observed error, as it considers the error of a slice that has been obtained using CLIP metadata instead of GT metadata. Or in other words, the observed error is the error of the DuT when tested on a slice of D that was obtained using the metadata information provided via CLIP (or in general some function G).
>
>
> ## Questions (Q)
>
> ### Q1: Section 5.1: What is the cutoff for the slice error?
>
> All experiments, with the exception of PanopticFCN, use a cutoff of $1.5 e_D$, where $e_D$ denotes the average error on the entire considered dataset and therefore differs across the experiments.
> For example, for celebA the ViT model has an accuracy of $94.48\%$ on the $202\,599$ images in the dataset. That means $5.52\%$ of the images were not classified correctly as person and misclassified as, e.g.,  "artifact" or "pompadour" even though the image contains a person. In this case, we therefore use a cutoff of $1.5 e_D \approx 8.3\%$. That is, we only consider a slice as a weak slice if its error rate is above this threshold.
>
> ### Q2: Section 5.1 Are the ODD the target attributes from the CelebA dataset?
>
> This is correct. The ODD for the celebA experiment is taken from target attributes from the dataset. The presence of ground-truth metadata makes celebA a good choice for testing metadata generation with CLIP. 40 attributes are available in CelebA ranging from attributes that map to dimensions from autonomous driving like gender, skin-color, age, etc to dimensions that are less important like pointy nose, bangs, wavy hair, etc. Therefore, for this work, we only choose the subset of attributes listed in Appendix A.3.
>
> ### Q3: Table 1: Was the complete CelebA dataset used as test-data for DuT? In any case, an inclusion of the portion of the slice to the actual data would be helpful.
>
> This is correct. As the ViT model used as DNN-under-test (DuT) for the celebA experiment is trained on imagenet21k, we could use the entire celebA dataset as a test set. We updated the caption in table 1 to include the full dataset size (see changes in blue).

---

> ### Author Response · Authors · 2025-04-30
> **Continuation of response to reviewer (3)**
>
> ### Q4: Table 2: How large is the overlap between the slices found with SWD-3 and the oracle?
>
> The methods we present work with statistical quantities describing slices (error rates, sizes). While for SWD-1, which is based fully on metadata annotation, these statistics are determined from actual sets, this is no longer the case for SWD-3. In the latter method, we try to correct the statistical quantities obtained from SWD-1 using knowledge on the labeling quality with bayesian noise correction. But these corrections cannot be linked to the underlying datasets. Consider, for example, $S_4$ in Table 1 on the CelebA experiment. In this case a GT slice with the attributes "wearing-hat=True" and "age=young" exists that contains $~7000$ samples. At the same time, using Clip on the data, we obtained a weakly labelled slice containing close to 13,000 samples (column for SWD-1). While both slices are actual subsets of the data and a meaningful overlap does exist, we already know from the performance of our labelling process that likely the SWD-1 slice is too large (and has a lower observed error). We therefore calculate estimated corrected values of these statistics (SWD-3), which are closer to the GT-slice properties. But, these corrected values do not directly correspond to a subset of the data. For instance, while we know that the slice should be smaller (from SWD-1 to SWD-3), we do not know exactly which elements should not be in the slice. So we can only correct or estimate the error and size at slice level but cannot make statements about the samples in it.
>
> ### Q5: How bad/weak are the actual predictions of the slice data?
> In the CelebA experiment, the weaknesses of a slice can be directly inferred from the error values. For example, in Table 1, Row 1, for SWD-1, we report $P(e|C) = 0.33$ with a slice size of 12,152. This indicates that, within the CLIP labelled slice, approximately $12,152 \times 0.33 \approx 4,010$ images have been misclassified by the DNN-under-test (DuT), specifically the ViT model. For comparison, defining the same semantic slice using GT labels instead, the error rate increases to $51\%$ while the slice size decreases to 6267 samples, implying about 3000 misclassifications.
> In Figure 3, we illustrate the most commonly occurring misclassifications. Thus, even for the relatively simple task of classifying the "person" class in the CelebA dataset, the model exhibits significant weaknesses.
> In the AD experiments, a similar approach is applied: the error estimate and slice size are used to determine the number of errors within each slice. In this case, since the task is object detection, semantic segmentation, or panoptic segmentation, an error is defined as a complete failure of the DuT to detect a pedestrian (IoU = 0).
>
> ## Requested Changes (RC)
>
> ### RC1: It would be helpful to provide a clearer explanation of why converting to structured data is superior to using approaches with unstructured data.
>
> We have updated Section 3 (see changes in blue) following W2's suggestions, where we compare SOTA SDMs (both slice-and-tag and tag-and-slice) using our notation. We highlight the importance of converting to structured data by discussing the limitations of slice-and-tag approaches, particularly their lack of adherence to a predefined specification $\mathcal{Z}$ when identifying weak slices. Because these approaches do not enforce a specification, the identified weaknesses may be neither actionable nor sufficiently granular, as evidenced by our results (Table 2 and Appendix C.2).
>
> ### RC2: A rewrite of the sections mentioned (in section 3) is needed for better clarity.
> We thank the reviewer for this suggestion and have updated section 3 based on the comments in W1, W2, W3, and CL1.
>
> ### RC3: A clearer description of the metadata would be helpful.
> As per W1, we have updated the metadata generation subsection with an example to clarify the workflow.
>
> ### RC4: How could the approach be adapted if no ODDs are provided? Could these be generated using an LLM?
>
> As discussed in W4, we have updated the Conclusion with a discussion regarding this (see changes in blue in Conclusion in "On generation of ODDs" ).

---

> ### Author Response · Authors · 2025-04-30
> **Continuation of response to reviewer (4)**
>
> ### RC5: A deeper discussion on how to handle false positive systematic error is missing.
> Thank you for this suggestion. As per W6, we have updated the conclusion to address this with a deeper discussion (see changes in blue in Conclusion in "On false positives".
>
> ### RC6: How can biases from pretrained models and foundation models be identified and potentially overcome?
>
> As per W7, we have addressed this concern for both using foundation models for metadata generation and considering foundation models themselves as DuT. We have updated the conclusion section with a discussion (see changes in blue in "On considering foundation models as \textbf{DuT}").
>
>
>
> ## Broader impact concerns (BIC)
>
> ### BIC 1: A deeper discussion on how to handle false positive systematic error is missing.
>
> As per W6 and RC5, we have updated the conclusion to address this with a more deeper discussion.
>
>
> ### BIC 2: How can biases from pretrained models and foundation models be identified and potentially overcome?
>
> See RC6.
>
>
> ## References
> - [1] - Burton, S., et al., (2022). Safety assurance of machine learning for perception functions. In Deep Neural Networks and Data for Automated Driving: Robustness, Uncertainty Quantification, and Insights Towards Safety (pp. 335-358). Cham: Springer International Publishing.
> - [2] - https://www.asam.net/standards/detail/openodd/
> - [3] - Chen, M., et al., (2025) HiBug2: Efficient and interpretable error slice discovery for comprehensive model debugging. In Proceedings of the Thirteenth International Conference on Learning Representations (ICLR 2025).
> - [4] DOMINO - Eyuboglu, S., et al., (2022). Domino: Discovering systematic errors with cross-modal embeddings. In Proceedings of the Tenth International Conference on Learning Representations (ICLR 2022).
> - [5] SVM-FD - Jain, S., et al., (2023). Distilling model failures as directions in latent space. In The Eleventh International Conference on Learning Representations.
> - [6] Gannamaneni, S. S., et al., (2023). Investigating CLIP Performance for Meta-Data Generation in AD Datasets. In Proceedings of the IEEE/CVF Conference on Computer Vision and Pattern Recognition (CVPR) Workshops (pp. 3840–3850).

---

> > ### Author Response · Authors · 2025-05-04
> > **Update on new (requested) experiments**
> >
> > As discussed previously, we have added appendix D and E as per requested experiments from reviewer CMrK.
> >
> > Regarding CL1, we have evaluated GPT-4o model on metadata generation quality and compared against CLIP on the relevant attributes in the newly added appendix E. As mentioned, the GPT model, due to improvements in training data quality, sheer size does outperform CLIP on multiple dimensions. However, CLIP obtains similar scores for two dimensions and outperforms GPT on further two dimensions. However, the main takeaway is that even the GPT model is not perfect and the Bayesian error correction proposed in our work would be needed to identify the correct weak slices. Additionally, the improvements in performance come at computational cost which might not scale for larger datasets.

---

### Review · Reviewer_wGv7 · 2025-04-18

**Summary Of Contributions:**

The paper presents a method for identifying systematic weaknesses in vision models by discovering underperforming data slices that align with human-understandable dimensions defined in the operational design domain (ODD). To achieve this, the authors combine zero-shot semantic labeling using CLIP with a modified version of the SliceLine algorithm for slice discovery. The proposed approach also includes a mechanism to handle noise in the generated metadata, improving robustness when CLIP labels are suboptimal. The method is validated on both synthetic and real-world datasets, and is used to analyze several pre-trained DNNs, revealing interpretable and relevant failure patterns.

**Audience:**

Yes

**Broader Impact Concerns:**

As a reviewer, I do not see any immediate ethical concerns that would necessitate a detailed Broader Impact Statement beyond what is typically expected. The proposed method focuses on analyzing model weaknesses along human-understandable dimensions, which could contribute positively to improving the safety and robustness of vision systems, particularly in safety-critical applications.

**Claims And Evidence:**

Yes

**Requested Changes:**

In Section 2, it would be helpful if the authors described the existing SDM methods using the same notations introduced in the proposed method (Section 3). This would make it easier to clearly understand the distinctions between prior approaches and the proposed DuT framework. Additionally, it would be interesting to know whether previous slice discovery methods have attempted to address the impact of noise from a Bayesian perspective. If such approaches exist, comparing them with the proposed Bayesian formulation for handling noisy metadata could strengthen the contribution and provide further insight. Furthermore, to enhance the practical impact of the work, the authors could consider incorporating methods to improve metadata quality, such as lightweight human-in-the-loop corrections or fine-tuning foundation models like CLIP on domain-specific datasets.

**Strengths And Weaknesses:**

The proposed approach offers several strengths. It identifies systematic weaknesses of deep vision models along human-understandable dimensions defined by the operational design domain (ODD), making the results interpretable and actionable. By leveraging CLIP for zero-shot image classification, the method avoids the need for manually annotated metadata, allowing it to scale efficiently to large image datasets. It also accounts for noisy or imperfect metadata, increasing robustness in practical settings.

---

> ### Author Response · Authors · 2025-04-27
> **Response to reviewer**
>
> We thank the reviewer for their time and feedback and for highlighting the strengths of our work. We address the mentioned weaknesses and requested changes below:
>
> ## 1. Unified description of SDM Methods
>
> Thank you for this valuable suggestion. We have updated section 3 paragraph 3 to reflect these changes (colored in blue). To simplify the exposition, we adopted terminology from the concurrently released HiBug2 [1] paper differentiating between slice-and-tag and tag-and-slice (our) methods. To describe the former methods, we introduced new notation for an embedding DNN denoted by $\mathcal E(s)$ (in practice, a CLIP model is also used for this task) using which the data is embedded and then clustering operations are performed to identify weak slices (e.g., like in DOMINO [2]).
> We then contrast these approaches with tag-and-slice methods like ours that first attempt to describe the semantic properties of $s$. For this step, approaches utilize models like CLIP for embedding the samples and performing a classification step to identify the semantic properties. Then search based algorithms are used to find for weak slices. The updated text contains a detailed comparison of SOTA SDMs using our notation.
> Furthermore, as requested by reviewer 56t6, we also included PromptAttack [4] in the comparison.
>
>
> ## 2. Related work on (Bayesian) noise correction
>
> To the best of our knowledge, previous work has not utilized a noise correction approach Bayesian or otherwise to address noise in the slice discovery process. Purely for structured data, SliceLine [5] proposes using minimum set size to ensure statistical significance in Slices. For images, approaches that perform slice-and-tag like DOMINO [2], Spotlight [6], SVM-FD [3] do not address noise in CLIP as semantics in $\mathcal Z$ occur only in the final step of tagging. PromptAttack and HiBug2 are two methods that perform tag-and-slice, where noise correction could be beneficial. However, these works do not provide ways to correct for the noise.
>
>
> ## 3. On improving CLIP performance
>
> Using either fine-tuning or human-in-the-loop corrections for metadata generation is a reasonable, and in practice, probably a required extension. We already mentioned both as potential directions for future work in our conclusion. Here, we would like to also point out that the Bayesian noise correction we propose would help identify dimensions where human-in-the-loop or other corrections would be most beneficial. For example, if precision for specific dimensions is lower than an acceptable level, e.g., to reach a desired test depth, only these dimensions could be corrected with human-in-the-loop intervention. As a remark, such granular control on slice quality would not be possible with slice-and-tag approaches as there is no (simple) metric to measure the quality of embeddings. Furthermore, as requested by reviewer CMrK, we will provide an update comparing the CLIP model against a GPT model for the celebA dataset.
>
> ## References
>
> - [1] Chen, M., et al., (2025). HiBug2: Efficient and Interpretable Error Slice Discovery for Comprehensive Model Debugging. In The Thirteenth International Conference on Learning Representations. Available at: https://openreview.net/forum?id=l30moNjSY9.
> - [2] DOMINO - Eyuboglu, S., et al., (2022). Domino: Discovering systematic errors with cross-modal embeddings. In Proceedings of the Tenth International Conference on Learning Representations (ICLR 2022).
> - [3] SVM-FD - Jain, S., et al., (2023). Distilling model failures as directions in latent space. In The Eleventh International Conference on Learning Representations.
> - [4] PromptAttack - Metzen, J.H. et al., 2023. Identification of Systematic Errors of Image Classifiers on Rare Subgroups. In Proceedings of the IEEE/CVF International Conference on Computer Vision (ICCV). pp. 5064–5073
> - [5] - Sagadeeva, S. and Boehm, M., 2021. SliceLine: Fast, Linear-Algebra-Based Slice Finding for ML Model Debugging. In Proceedings of the 2021 International Conference on Management of Data. SIGMOD ’21. Virtual Event, China: Association for Computing Machinery, pp. 2290–2299. Available at: https://doi.org/10.1145/3448016.3457323.
> - [6] Spotlight - D’ Eon, G. et al., 2022. The spotlight: A general method for discovering systematic errors in deep learning models. In 2022 ACM Conference on Fairness, Accountability, and Transparency. pp. 1962–1981.

---

> ### Author Response · Authors · 2025-05-04
> **Update on new (requested) experiments**
>
> As discussed previously, we have added appendix D and E as per requested experiments from reviewer CMrK.
>
> Regarding 3, we have evaluated GPT-4o model on metadata generation quality and compared against CLIP on the relevant attributes in the newly added appendix E. As mentioned, the GPT model, due to improvements in training data quality, sheer size does outperform CLIP on multiple dimensions. However, CLIP obtains similar scores for two dimensions and outperforms GPT on further two dimensions. However, the main takeaway is that even the GPT model is not perfect and the Bayesian error correction proposed in our work would be needed to identify the correct weak slices. Additionally, the improvements in performance come at computational cost which might not scale for larger datasets.

---

### Review · Reviewer_CMrK · 2025-04-20

**Summary Of Contributions:**

The authors address the problem of systematically finding weaknesses of DNNs by using foundation models with a novel algorithm to annotate metadata. The approach is particularly useful in dealing with noise in data annotation and does not require extensive human efforts. To validate their approach, they conduct experiments on both synthetic and realistic CV datasets.

**Audience:**

Yes

**Broader Impact Concerns:**

I do not have any concern.

**Claims And Evidence:**

Yes

**Requested Changes:**

1. Address whether new models/datasets can be used.

2. Discuss about scaling possibilty of the proposed method.

3. Discuss more about why CLIP is used for annotaion.

**Strengths And Weaknesses:**

Strengths:

1. The addressed problem is important: to provide more interpretability for DNNs and ensure their safety.

2. The paper writing is easy to follow: the storyline is very clear, and the presentation is good.

Weaknesses:

1. Evaluated models and datasets are not state-of-the-art: the models evaluated are from 2015 to 2021, and datasets are from 2015 to 2020 (page 7). Can newer models and datasets also be considered?

2. Scaling possibility can be discussed: Following weakness 1, how good is the proposed method in terms of scaling to very large models and very large datasets?

3. Why CLIP for annotation: The authors mention that "CLIP achieved SOTA level zero- shot performance on different dimensions (page 5)", the reference points to Gannamaneni et al. (2023). If I understand it correctly, the SOTA results are by 2023. Is CLIP still SOTA now? If not, do alternatives exist?

---

> ### Author Response · Authors · 2025-04-27
> **Response to reviewer**
>
> We thank the reviewer for their time and helpful feedback. We address the requested changes and mentioned weaknesses below:
>
>
> ## 1. Addressing models and datasets
>
> Regarding the models, although the main model architectures were proposed between 2015--2021, we have chosen purposefully publicly available model weights from popular and mature object detection and segmentation evaluation repositories.
>
> For example, if we consider the Faster-RCNN model used in our AD experiment, the backbone used is **ConvNeXt-T** with lr scheduler - 1x (see Appendix A.1, paragraph 2), released in 2022. This model is also one of the final additions to the Model Zoo of the BDD100K Dataset given on the respective repository [1]. On this repository, the model offers a competitive baseline.
>
> Similarly, for Cityscapes, the considered **SeTR-PUP** model---while proposed in 2020---has been trained to achieve an mIoU of 79.21 [2]. The most recent 2023 model, **DDRNet**, evaluated on Cityscapes on the same repository, reports an mIoU of 79.99 [3]. Since the performance difference between both models is very minimal for this dataset, the SeTR model remains competitive.
>
> As for datasets, while they have been released between 2015-2020, Cityscapes and BDD100k continue to remain popular benchmark datasets for Autonomous driving evaluations and CelebA for slice discovery method testing (e.g., in DOMINO [4] or SVM-FD [5]). Furthermore, we believe that if models display systematic weaknesses on these older datasets themselves, these issues are still worth evaluating before evaluating models on newer datasets.
>
> To further address the reviewers concerns, including the question on scalability below, we will include a further analysis of a more recently released and larger model (likely Yolov11M or similar) on the EuroCity Persons Dataset [6]. This dataset contains substantially more pedestrian annotations (~200k) than in the datasets we considered previously addressing the scalability issue in datasets. We will provide the update in coming days.
>
> ## 2. Details on Scalability.
>
> As mentioned above, we are evaluating a larger model and a larger dataset to demonstrate scalability w.r.t. model and dataset sizes.
> To give some further insight w.r.t. the impact of metadata dimensions in Z and the number of semantic combinations we want in our weaknesses, we have added a subsection in appendix regarding scalability (see appendix A.5). All changes in the paper are marked in blue.
> In summary, while the search for weak slices does scale non-linearly with the search depth and therefore increases algebraically with Z, the main computational effort in our experiments is the generation of metadata, which scales linearly with  | D | and | Z | and inference time with the DuT.

---

> > ### Author Response · Authors · 2025-04-27
> > **Continuation of the response to reviewer**
> >
> > ## 3. Why CLIP for annotation.
> > Our usage of CLIP is due to its lightweight nature and prevalent usage in existing Slice discovery methods, such as DOMINO [4] and SVM-FD [5]. However, recent works like HiBug2 [7] have used powerful GPT models for the metadata generation step. While these models are expected to perform due to their sheer size and generality, we do not expect such solutions to scale in terms of cost. However, we will perform and update the paper with a comparison of CLIP and GPT on the CelebA dataset in a similar experiment setup to Gannamaneni et al. [8]. Details will be given in the appendix. We will provide the update in coming days.
> >
> > Further, one of the key points of our work is to demonstrate how slice discovery can work even if the deployed model for annotation, e.g., CLIP, produces noisy meta-data labelling. To illustrate this on a "toy-dataset" such as CelebA the use of a "weaker" model is beneficial. In practice, for more challenging scenarios, one might expect that also complex models such as GPT or variants might become error prone (see also "fine-tuning" in the answer to reviewer wGv7), but demonstrating the effects without proper GT labels would be cumbersome.
> >
> > ## References
> > - [1] https://github.com/SysCV/bdd100k-models/tree/main/det - BDD100K Model Zoo
> > - [2] https://github.com/open-mmlab/mmsegmentation/tree/main/configs/setr - SeTR-PUP Configuration
> > - [3] https://github.com/open-mmlab/mmsegmentation/tree/main/configs/ddrnet - DDRNet Configuration
> > - [4] DOMINO - Eyuboglu, S., et al., (2022). Domino: Discovering systematic errors with cross-modal embeddings. In Proceedings of the Tenth International Conference on Learning Representations (ICLR 2022).
> > - [5] SVM-FD - Jain, S., et al., (2023). Distilling model failures as directions in latent space. In The Eleventh International Conference on Learning Representations.
> > - [6] https://eurocity-dataset.tudelft.nl/eval/overview/statistics
> > - [7] Chen, M., et al., (2025). HiBug2: Efficient and Interpretable Error Slice Discovery for Comprehensive Model Debugging. In The Thirteenth International Conference on Learning Representations. Available at: https://openreview.net/forum?id=l30moNjSY9.
> > - [8] Gannamaneni, S. S., et al., (2023). Investigating CLIP Performance for Meta-Data Generation in AD Datasets. In Proceedings of the IEEE/CVF Conference on Computer Vision and Pattern Recognition (CVPR) Workshops (pp. 3840–3850).

---

> > > ### Author Response · Authors · 2025-05-04
> > > **Update on requested experiments**
> > >
> > > We have now updated the appendix with additional experiments to address points 1 and 3.
> > >
> > > Regarding 1, we have evaluated a publicly available YOLOv11m (released in 2024) model on the comparatively larger Eurocity Persons dataset. As the dataset contains day and night subsets, we perform the experiment on both subsets separately with our algorithm and present the results of human evaluation of metadata generation with CLIP, identified weak slices and slice descriptions in the newly added appendix D.
> > >
> > > Regarding 3, we have evaluated GPT-4o model on metadata generation quality and compared against CLIP on the relevant attributes in the newly added appendix E. As mentioned, the GPT model, due to improvements in training data quality, sheer size does outperform CLIP on multiple dimensions. However, CLIP obtains similar scores for two dimensions and outperforms GPT on further two dimensions. However, the main takeaway is that even the GPT model is not perfect and the Bayesian error correction proposed in our work would be needed to identify the correct weak slices. Additionally, the improvements in performance come at computational cost which might not scale for larger datasets.

---

### Author Response · Authors · 2025-07-07
**Thank you**

We thank the three reviewers and the action editor for their valuable time and feedback and for the decision. We have now uploaded the camera-ready version.

---

### Decision · Action_Editor_MmRw · 2025-06-06

**Recommendation:** Accept as is

**Audience:**

Yes

**Audience Explanation:**

Analyzing potential weaknesses in black-box vision models represents a crucial objective in explainable AI research. It has potentially far-reaching implications across various AI applications.

**Claims And Evidence:**

Yes

**Claims Explanation:**

This paper proposes an approach to detecting and improving weaknesses in vision models. The proposed method is practically valuable for safety-critical applications. By combining foundation models for metadata generation with an enhanced slice discovery algorithm and Bayesian-inspired noise correction, the authors developed a new framework that identifies systematic model vulnerabilities in human-interpretable ways. The method's effectiveness was demonstrated through comprehensive validation on both synthetic and real-world datasets across various pre-trained DNNs, and it yields actionable insights into model behavior. The authors' thorough rebuttal successfully addressed reviewer concerns, improving the paper's clarity and handling bias-related issues. Overall, the work was praised for its practical contribution to understanding and enhancing vision models in safety-sensitive domains.